# Sortase A Fusion Expression and mIFc2 Co-Expression of Bovine Lactoferricin and Analysis of Its Antibacterial Activity

Chao-Yu Hsu [1,2,†], Chung-Yiu Hsieh [3,†], Cheng-Yao Yang [4], Yu-Kang Chang [5,6], Wen-Ling Shih [7], Chuan-Ming Yeh [3,8], Nien-Jen Hu [9], Ming-Shan Chen [10], Brent L. Nielsen [11] and Hung-Jen Liu [2,3,12,13,14,*]

1   Division of Urology, Department of Surgery, Tungs' Taichung MetroHarbor Hospital, Taichung 402, Taiwan
2   Ph.D. Program in Translational Medicine, National Chung Hsing University, Taichung 402, Taiwan
3   Institute of Molecular Biology, National Chung Hsing University, Taichung 402, Taiwan
4   Graduate Institute of Veterinary Pathobiology, National Chung Hsing University, Taichung 402, Taiwan
5   Department of Medical Research, Tungs' Taichung MetroHarbor Hospital, Taichung 402, Taiwan
6   Department of Post-Baccalaureate Medicine, National Chung Hsing University, Taichung 402, Taiwan
7   Department of Biological Science and Technology, National Pingtung University of Science and Technology, Pingtung 900011, Taiwan
8   Bioproduction Research Institute, National Institute of Advanced Industrial Science and Technology, Tsukaba 300-4352, Japan
9   Gradute Institute of Biochemistry, National Chung Hsing University, Taichung 402, Taiwan
10  Department of Anesthesiology, Ditmanson Medical Foundation Chia-Yi Christian Hospital, Chia-Yi 600040, Taiwan
11  Department of Microbiology and Molecular Biology, Brigham Young University, Provo, UT 4007, USA
12  The iEGG and Animal Biotechnology Center, National Chung Hsing University, Taichung 402, Taiwan
13  Rong Hsing Research Center for Translational Medicine, National Chung Hsing University, Taichung 402, Taiwan
14  Department of Life Sciences, National Chung Hsing University, Taichung 402, Taiwan
*   Correspondence: hjliu5257@nchu.edu.tw; Tel.: +886-4-22840485 (ext. 243); Fax: +886-4-22874879
†   These authors contributed equally to this work.

**Abstract:** The coding region for the sortase A (SrtA) of *Staphylococcus aureus* was fused at the N-terminus of LfcinB. The SrtA-LfcinB fusion protein in *E. coli* C43(DE3) was expressed with the expected sizes of 21 kDa and 38 kDa by pET21b-SrtA-LfcinB and pET32-1SrtA-LfcinB constructs, respectively. Increased levels of the TrxA-His-SrtA-SrtA-LfcinB fusion protein were detected by the pET32-3SrtA-LfcinB construct having three expression cassettes. LfcinB is released from the expressed SrtA-LfcinB protein by SrtA self-cleavage which is induced in the presence of $Ca^{2+}$. The antibacterial activity was detected after SrtA-mediated cleavage of LfcinB. Furthermore, to reduce the antimicrobial peptide toxicity to the *E. coli* host, the human interferon-γ (hIFN-γ) sequences were mutated into a negatively charged mIFc2 protein (7 kDa), which was co-expressed with LfcinB in an insoluble form. The yield of LfcinB was elevated while changing the gene order of LfcinB and mIFc2 (pET21b-fLfcinB-bmIFc2). Furthermore, increased levels of LfcinB were detected using the pET21b-(fLfcinB-bmIFc2)$_2$ construct. To increase the dissolution rate of inclusion bodies, inclusion bodies treated with different temperatures and pH and resuspended in different volumes of 50 mM Tris-HCl were assayed. Our results reveal that heat-treated LfcinB/mIFc2 inclusion bodies at 90 °C, pH 10, and 16X resuspended volumes have the best resolubilization rate. This work suggests that the mIFc2 co-expression system shows higher efficiency for LfcinB production than the SrtA fusion system. The expressed LfcinB from the mIFc2 co-expression system exhibits excellent broad-spectrum antibacterial activities against thirteen Gram-negative and ten Gram-positive bacteria species with a range of minimum inhibitory concentrations (MIC) between 37–150 ug/mL.

**Keywords:** antimicrobial peptides; lactoferricin B; staphylococcus aureus transferase SrtA fusion expression; mIFc2 co-expression

## 1. Introduction

In the past, to maintain the health of animals and to prevent infectious diseases, a variety of antibiotics were added to animal feed [1–5]. Although the addition of antibiotics in addition to reducing the growth of bacteria can promote the meat production rate of livestock, excessive addition of antibiotics also causes the problem of antibiotic abuse. Overuse of antibiotics and their residues in the environment increases bacterial resistance in humans and farmed animals [6–9]. Since this vicious cycle forces humans to increase the dosage to kill bacteria, there may even be a dilemma that no effective antibiotics are available when bacterial infections occur again in the future. Recently, the European Union's new laws came into force, banning farmed animals from being routinely fed a diet of antibiotics. Therefore, finding an emerging antibacterial substance to replace antibiotics has become a most urgent issue. Antimicrobial peptides (AMPs) are an integral part of the innate immune system that protects a host from invading pathogenic bacteria [10]. They are widely distributed in various animals and plants [11–14]. AMPs are generally composed of less than 50 amino acids and are characterized by cationic amphipathic properties, which show broad-spectrum antimicrobial activities against various microorganisms, including Gram-positive and Gram-negative bacteria, fungi, and viruses [10]. Their antibacterial mechanism is completely different from traditional antibiotics. Of particular interest, many AMPs are effective against multi-drug resistant (MDR) bacteria and possess low propensity for developing resistance [15–17].

To date, the two most studied lactoferricin (Lfcin) molecules are human and bovine (LfcinH and LfcinB) [18,19]. Although both peptides are highly positively charged, there is a striking difference in their length and amino acid sequences [18,20]. LfcinB is a tryptophan-rich, positively charged small molecule antibacterial peptide (25 amino acid residues; residues 17–41 of bovine lactoferrin, bLF), which is an iron-binding protein bLF released from the N-terminus by pepsin hydrolysis in the acidic environment of the digestive tract [18,20]. In aqueous solution, LfcinB is changed from the original α-helical structure in bLF to a twisted anti-parallel β-sheet structure [21,22]. The first to thirteenth amino acids in the LfcinB form an α-helix, the twelfth to fifteenth amino acids form a turn, and the subsequent amino acids form a β-sheet. There is a disulfide bond between the thirteenth and twentieth cysteine, among which a hydrogen bond can stabilize the LfcinB structure [23]. Studies have confirmed that the antibacterial effect of LfcinB is better than that of bLF [18,20,24]. The anti-bacterial activity of bLF decreases in an environment with abundant iron, but the antibacterial effect of LfcinB is not affected by ions, and the antibacterial effect is 400 times that of bLF [18,20,24]. In comparing the antibacterial effects of lactoferrin in cattle, humans, mice, and pigs (indicated by LfcinB, LfcinH, LfcinM, and LfcinP), LfcinB has the best antibacterial effect [18,20,24].

There are three major methods for producing antimicrobial peptides, such as extraction from organisms, chemical synthesis, and expression via genetic engineering [25–32]. Most of the production of antimicrobial peptides uses a genetic engineering approach. The protein yield obtained by using an *E. coli* expression system is better due to the lower cost of using a bacterial host. However, due to the toxicity of the expressed AMP to the host, it is difficult to use *E. coli* to express antimicrobial substances. To overcome this problem, two different expression approaches were established to express LfcinB. The aim of this study is to reduce the antimicrobial peptide toxicity to the *E. coli* host and to increase expression levels of LfcinB. In this work, SrtA fusion expression and mIFc2 co-expression have been developed. After expression of this SrtA-LfcinB fusion protein, the enzyme activity of SrtA can be used to induce self-cleavage and release of LfcinB in the presence of $Ca^{2+}$ [30]. The SrtA-LfcinB fusion protein neutralizes and reduces the toxicity of LfcinB. In the mIFc2 co-expression system, the method relies on a translationally coupled two-cistron system, in which the termination codon for the first cistron (which encodes the anionic polypeptide mIFc2, a derivative of hIFN-γ) overlaps with the initiation codon for the second cistron (which encodes a cationic LfcinB) [32]. The negative charge of mIFc2 neutralizes the positive charge of LfcinB to reduce its toxicity to *E. coli* [32]. In comparison with these

two expression strategies, we found that the mIFc2 co-expression system showed higher efficiency for LfcinB production than that of the SrtA fusion system. The expressed LfcinB produced by the mIFc2 co-expression system exhibits antibacterial activity against thirteen Gram-negative and ten Gram-positive bacteria species with MIC between 37–150 ug/mL.

## 2. Materials and Methods

### 2.1. E. coli Host Cells and Gram-Negative and Gram-Positive Bacteria Species

In this work, thirteen Gram-negative and ten Gram-positive bacteria species (Table 1) (Food Industry Research and Development Institute, Hsinchu, Taiwan) were used to test antibacterial activities of LfcinB. In this study, five different *Escherichia coli* strains were tested, including BL21(DE3), BL21(DE3)-pLysS, Origami(DE3)-pLysS, C41(DE3), and C43(DE3) (Food Industry Research and Development Institute).

**Table 1.** Minimum inhibitory concentrations of LfcinB determined using microbroth dilution method.

| Microorganisms | LfcinB ug/mL |
| --- | --- |
| **Gram negative bacteria** | |
| *Escherichia coli* 10675 | 100 |
| *Salmonella choleraesuis* | 37 |
| *Salmonella typhimurium* | 84 |
| *Klebsiella oxytoca* 13985 | 62 |
| *Enterobacter aerogenes* 10370 | 62 |
| *Aeromonas hydrophila* Ah | 37 |
| *Vibrio alginolyticus* 12829 | 75 |
| *Listonella anguillarum* 13810 | 37 |
| *Gardnerella vaginalis* 17040 | 75 |
| *Yersinia enterocolitica* subsp. 13999 | 75 |
| *Pseudomonas aerruginosa* | 150 |
| *Vibrio parahaemolyticus* | 75 |
| *Vibrio vulnificus* | 75 |
| **Gram positive bacteria** | |
| *Bacillus subtilis* | 75 |
| *Micrococcus luteus* 11034 | 75 |
| *Streptococcus agalactiae* 10787 | 37 |
| *Listeria monocytogenes* 14845 | 37 |
| *Streptococcus pyogenes Rosenbach* 10797 | 37 |
| *Staphylococcus aureus* 10451 | 75 |
| *Staphylococcus* sp. 10783 | 75 |
| *Streptococcus pneumoniae* 10794 | 75 |
| *Staphylococcus haemolyticus* 15237 | 75 |
| *Enterococcus faecalis* 10066 | 62 |

### 2.2. Amplification of LfcinB, SrtA Region of Staphylococcus aureus Transferase, and SrtA-LfcinB Coding Fragments

To express the SrtA-LfcinB fusion protein, polymerase chain reaction (PCR) and overlapping PCR were used to amplify the SrtA domain of *Staphylococcus aureus* transferase. To amplify the LfcinB coding region, primer pairs were chosen according to the sequences of the mature peptide of bovine lactoferricin B [24]. Restriction sites in the primers were designed according to the requirements of the expression vectors. To amplify the sequences of the SrtA active region, primer pairs were designed according to the sequences of the SrtA active region of *Staphylococcus aureus* [30]. The SrtA active region was amplified from *Staphylococcus aureus.* For co-expression of mIFc2 and LfcinB, primer pairs were designed according to the sequences of the hIFN-γ gene. To create the mIFc2 gene [32], six positively charged amino acids (arginine and lysine) in the hIFN-γ gene were mutated to negatively charged amino acids (aspartates and glutamates). Purified PCR products were digested with the respective enzymes, followed by ligation into the respective vectors. All constructs were confirmed by restriction enzyme digestion and DNA sequencing. All primers used



in this study and the positions of restriction sites designed in the primers are shown in Supplementary Table S1.

### 2.3. Construction of the pET32a Vector Carrying Multiple Expression Cassettes to Express the SrtA-LfcinB Fusion Protein and Development of LfcinB and mIFc2 Co-Overexpression Vectors

In this study, the SrtA gene with a SrtA recognition cutting site at the 3′end and a BamHI restriction site at the 5′end was amplified by PCR. PCR products were purified and subcloned into the yT&A cloning vector (Yeastern Biotech, Taipei, Taiwan) and the resultant plasmid was named pT-SrtA. Next, the LfcinB gene with a SrtA recognition cutting site at the 5′end and a BamHI restriction site at the 3′end was amplified by PCR. A diagram showing the SrtA-LfcinB fusion gene with a 5′end NdeI and 3′end SalI restriction sites is shown in Supplementary Figure S1A. This LfcinB gene fragment was amplified by overlapping PCR using primers SrtA-F1 and LfcinB-R1. PCR products were subcloned into the yT&A cloning vector and the resultant plasmid was named pT-SrtA-LfcinB. pET32a-SrtA-LfcinB and pET21b-SrtA-LfcinB constructs were also constructed. A diagram showing the construction steps of the pET32a vector (EMD Biosciences Inc., San Diego, CA, USA) with multiple (two and three) expression cassettes are shown in Supplementary Figure S1B. To amplify the full-length mIFc2 DNA fragment (211 bp), overlapping PCR was performed using the respective primers as shown in Supplementary Figure S2A. The 3′end of mIFc2 contains the same complementary fragment as the 5′end of the DNA fragment of LfcinB. Diagrams depicting the detailed steps for constructing LfcinB and mIFc2 co-expression vectors pT-mIFc2-LfcinB, pET21b-bmIFc2-fLfcinB, pET21b-fLfcinB-bmIFc2, and pET21b-(fLfcinB-bmIFc2)$_2$ are shown in Supplementary Figure S2B–D. All developed vectors were confirmed by Sanger sequence.

### 2.4. Expression of the SrtA-LfcinB Fusion Protein and Co-Expression of LfcinB and mIFc2

Recombinant plasmids were transformed into *Escherichia coli* host cells. Transformed *E. coli* cells were grown in Luria-Bertani broth with 100 μg/mL ampicillin at 37 °C to an optical density of 0.6 and then induced with 200 nM isopropyl β-D-1-thiogalactopyranoside (IPTG) for 3 h at 30 °C. After induction, cells were harvested by centrifugation, followed by resuspension in pET system lysis buffer (20 mM Tris-HCl, pH 8.0, 300 mM NaCl, 0.2 mM PMSF, 10% glycerol, 5 mM imidazole) and sonicated. The cell suspension was centrifuged at 12,000× *g* for 20 min at 4 °C. Expressed proteins were analyzed by a 16.5% tricine SDS-PAGE (Sigma-Aldrich, St. Louis, MO, USA).

### 2.5. Protein Purification and Self-Cleavage of the SrtA-LfcinB Fusion Protein to Release LfcinB

One hundred microliters of the 200 nM IPTG-induced bacterial solution was centrifuged, the pellet was washed with ddH$_2$O, and the bacterial cells were suspended in 10 ml of buffer A (20 mM Tris-HCl, pH 7.5, 50 mM NaCl, and 5 mM 2-mercaptoethanol [2-ME]). Bacteria were disrupted by sonication, and after centrifugation, the supernatant was filtered through a 0.45 M filter membrane to obtain a crude extract. The crude extract was placed in a Nickel affinity column followed by two rounds of washing with 10 times the column volume buffer (20 mM Tris-HCl, pH 7.5, 500 mM NaCl, 30 mM imidazole, and 5 mM 2-ME) to obtain the target protein. To perform induction of self-cleavage and release of LfcinB, cleavage buffer (20 mM Tri-HCl, pH 7.5, 50 mM NaCl, 5 mM 2-ME, 5 mM CaCl$_2$, and 5 mM Gly3) was added at 25 °C for 4–6 h and then the waste solution was removed followed by adding 10 ml of elution buffer (20 mM Tris-HCl, pH 7.5, 50 mM NaCl, 500 mM imidazole, and 5 mM 2-ME) to obtain the target protein. All chemicals and reagents used in this study were from Sigma-Aldrich.

### 2.6. SDS-Polyacrylamide Gel Analysis and Western Blot Assays

The concentrations of solubilized protein in the cell lysates were determined with the Bio-Rad protein assay (Bio-Rad, Hercules, CA, USA) according to the manufacturer's protocol. After centrifugation, the pellets were washed with ddH$_2$O, and the bacteria were

suspended in 100 uL of buffer containing 20 mM Tris-HCl, pH 7.5, 50 mM NaCl, and 5 mM 2-mercaptoethanol [2-ME] and disrupted by a sonicator (New Taiwan Fishery Co., Taichung, Taiwan). Equal amounts of samples were mixed with 2.5X Laemmli loading buffer (with 0.5% 2-ME) and boiled for 10 min in a water bath. The samples were electrophoresed in a 16.5% Tricine or 12% sodium dodecyl sulphate (SDS)-polyacrylamide gel and transferred to PVDF membrane (GE Healthcare Life Sciences, Chicago, IL, USA). Protein expression was examined using anti-His-tag antibodies (dilution 1:3000, Santa Cruz, Dallas, TX, USA) and horseradish peroxidase (HRP)-conjugated goat anti-mouse IgG (H + L) secondary antibody (dilution 1:5000; SeraCare, Milford, CT, USA). The results were detected on X-ray films (Kodak, Rochester, NY, USA) after membrane incubation with enhanced chemiluminescence (ECL plus) reagent (Amersham Biosciences, Little Chalfont, UK).

### 2.7. Treatments of Inclusion Bodies with Different Temperatures and pHs

To increase the dissolution rate of inclusion bodies, inclusion bodies were resuspended in different volumes of 50 mM Tris-HCl at different temperatures and pHs. Inclusion bodies were collected and resuspended in 50 mM Tris-HCl (pH 8) followed by heat treatment in a water bath for 15 min at different temperatures (60 °C, 70 °C, 80 °C, and 90 °C) Heat-treated LfcinB/mIFc2 inclusion bodies were analyzed on a 16.5% Tricine-SDS-PAGE gel. To study the effect of pH on inclusion body dissolution, the inclusion bodies were resuspended in 50 mM Tris-HCl at different pH (8, 8.5, 9, 9.5, and 10) heated at 90 °C for 15 min in a water bath, and the dissolution rate was quantified by the BioRad protein assay. The inclusion bodies were also resuspended in different volumes (1X, 2X, 4X, 6X, 8X, 10X, and 16X) of 50 mM Tris-HCl (pH 8) and analyzed as above.

### 2.8. Antibacterial Test Using the Agar Diffusion Method

The agar diffusion method is used to determine the lowest concentration of the assayed antimicrobial agent that, under defined test conditions, inhibits the visible growth of the bacterium being investigated. This method was used to determine the antimicrobial activity of LfcinB as described previously [33]. Briefly, the process includes the inoculation of bacterial cells on nutrient agar petri dishes and tested samples are overlaid. Afterward, the dishes are incubated for 24 h at 37 °C and a bacteriostatic circle was examined. Two-fold dilutions ranging between 12.5–200 mg/mL of SrtA-LfcinB were prepared and their antimicrobial activities were assessed. Fifty millimolar Tris-HCl (pH 8) was used as a negative control. The test was conducted in triplicates and zones of growth inhibition were recorded. Thirty μL of the test microorganisms prepared from 18 h broth cultures (diluted to 0.5 McFarland standard) were dispensed into the flat plate wells and incubated at 37 °C for 24 h. Bacterial growth in the wells was assessed after addition of 20 μL of 0.25 mg/mL 3-(4,5-dimethyl-2-thiazolyl)-2,5-diphenyl-2H-tetrazolium bromide (MTT) (Applichem Chemical Synthesis Services, Germany). The plates were incubated for 30 min, after which the wells were observed for any color changes. The well containing the least concentration that did not show any color change (a change from yellow to purple) was noted as the MIC of the extract.

### 2.9. Statistical Analysis

The Student t-test was used to evaluate the statistical significance of all data and all data were analyzed using SPSS software (IBM, New York, NY, USA). Each value represents mean $\pm$ standard error (SE) of three independent experiments. $p$ values of less than 0.05 were considered statistically significant.

## 3. Results

### 3.1. Development of Two Different Expression Systems for Expression of LfcinB

To reduce LfcinB toxicity to *E. coli* host cells and to increase expression levels of LfcinB, two different expression systems for LfcinB were established and they are shown in Figure 1A,B. The transferase active region SrtA of *Staphylococcus aureus* was fused with

LfcinB (Figure 1A). After expression of this SrtA-LfcinB fusion protein, the enzyme activity of SrtA can be used to induce self-cleavage and release of LfcinB [30]. Diagrams of the LfcinB-mIFc2 two-cistron co-expression system are shown in Figure 1B.

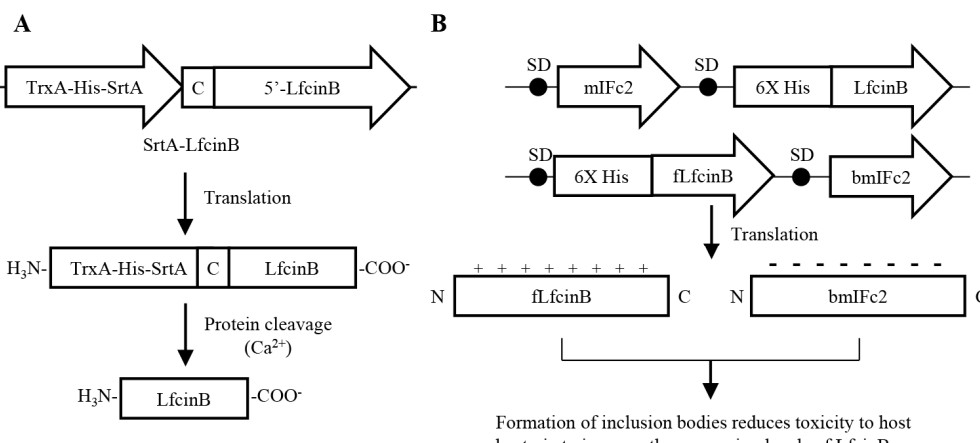

**Figure 1.** Two different expression systems for expression of LfcinB. (**A**). The transferase active region SrtA of *Staphylococcus aureus* was fused with LfcinB. After expression of this SrtA-LfcinB fusion protein, the enzyme activity of SrtA can be used to induce self-cleavage and release of LfcinB in presence of $Ca^{2+}$. (**B**) Diagrams of the LfcinB two-cistron co-expression system. The two different gene orders of LfcinB and mIFc2 are shown at top of the panel.

### 3.2. Amplification of SrtA, LfcinB, SrtA-LfcinB, and the Full-Length mIFc2 DNA Fragment As Well As Construction of pET21b-SrtA-LfcinB and pET32a-SrtA-LfcinB and the LfcinB and mIFc2 Co-Expression Vectors

In this study, the SrtA gene with a SrtA recognition cut site at the 3′end and a BamHI restriction site at the 5′end was amplified by PCR from Staphylococcus aureus. The expected size of pCR products is 474 bp in length (Figure 2A, left panel). PCR products were purified and subcloned into the yT&A cloning vector and the resultant plasmid was named pT-SrtA. Next, the LfcinB gene with a SrtA recognition cutting site at the 5′end and a BamHI restriction site at the 3′end was amplified by overlapping PCR. The expected size of PCR products is 92 bp in length (Figure 2A, right panel). PCR products were subcloned into the yT&A cloning vector. The resultant plasmid was named pT-LfcinB. LfcinB and SrtA DNA fragments were used as templates and the SrtA-LfcinB fusion gene with a 5′end NdeI and 3′end SalI restriction sites was amplified by overlapping PCR using primers SrtA-F1 and LfcinB-R1. The PCR product (546 bp) was amplified (Figure 2B) and subcloned into the yT&A cloning vector and the resultant plasmid was named pT-SrtA-LfcinB. To create pET32a-SrtA-LfcinB and pET21b-SrtA-LfcinB, the pT-SrtA-LfcinB was digested with NdeI and SalI and the resultant fragment was subcloned into the pET32a and pET21b vectors (EMD Biosciences Inc.), respectively. The resultant constructs were digested with restriction enzymes NdeI and SalI (New England Biolabs, Rowley, MA, USA) and the expected fragment (546 bp) was obtained (Figure 2C). To construct pET32a-(SrtA-LfcinB)$_2$, the pT-32rSrtA-LfcinB plasmid was digested with SpeI and SwaI (New England Biolabs) and the resultant fragment was subcloned into the NheI and SwaI restriction sites of the pET32a-1SrtA-LfcinB construct. The resultant plasmid was named pET32a-(SrtA-LfcinB)$_2$ (Supplementary Figure S1B). To construct the pET32a-(SrtA-LfcinB)$_3$ plasmid, the pT32-rSrtA-LfcinB plasmid was digested with SpeI and SwaI and the resultant fragment was subcloned into the NheI and SwaI restriction sites of the pET32a-2SrtA-LfcinB plasmid. The resultant plasmid was named pET32a-(SrtA-LfcinB)$_3$ (Supplementary Figure S1B). The maps of pET32a-1SrtA-LfcinB, pET32a-(SrtA-LfcinB)$_2$, and pET32a-(SrtA-LfcinB)$_3$ plasmids are shown in Supplementary Figure S3A. These constructs were digested with SwaI and PstI (New England Biolabs) and the expected fragments were obtained (Supplementary Figure S3B).

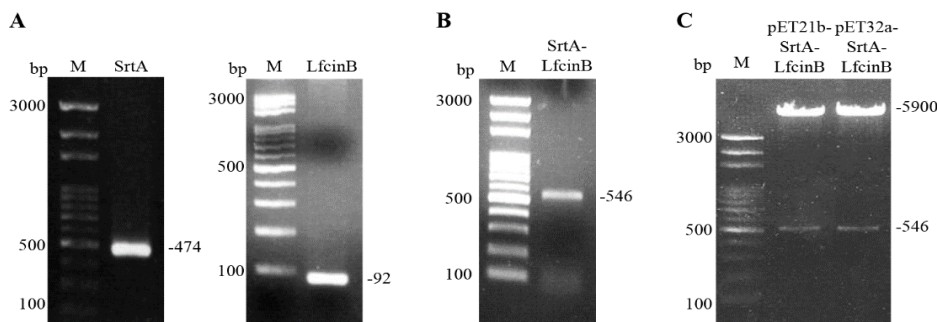

**Figure 2.** Amplification of the SrtA and LfcinB fusion gene and construction of pET32a-SrtA-LfcinB and pET21b-SrtA-LfcinB plasmids. (**A**)The SrtA gene with a SrtA recognition cuting site at the 3′end and a BamHI restriction site at the 5′end was amplified by PCR from Staphylococcus aureus. The expected size of the PCR products is 474 bp in length (left panel). (**B**) The LfcinB gene with a SrtA recognition cutting site at the 5′end and a BamHI restriction site at the 3′end was amplified by overlapping PCR. The expected size of the PCR products is 92 bp in length (right panel). (**B**) LfcinB and SrtA DNA fragments were used as templates and the SrtA-LfcinB fusion gene with a 5′end NdeI and 3′end SalI restriction sites was amplified by overlapping PCR. The PCR product (546 bp) was amplified. (**C**) pET32a-SrtA-LfcinB and pET21b-SrtA-LfcinB were digested with NdeI and SalI and the expected fragment (546 bp) was obtained. Lane M: DNA marker.

As shown in Figure 3A, the full-length mIFc2 DNA fragment (211 bp) was amplified by overlapping PCR. To amplify the front mIFc2 DNA fragment (90 bp), overlapping PCR was performed using primers mIFDNA1-F and mIFDNA1-R (Supplementary Figure S2A). By using primers mIFDNA2-F and mIFDNA2-R, the rear mIFc2 DNA fragment (105 bp) was amplified (Figure 3A). By using the front and rear mIFc2 DNA fragments as templates, the full-length mIFc2 DNA fragment (211 bp) was amplified (Figure 3A). The 3′end of mIFc2 contains the same complementary fragment as the 5′end of the LfcinB DNA fragment. Using pT-LfcinB as a template, the DNA fragment of LfcinB (140 bp) was amplified using primers mIFNHLfcinB-F and LfcinB-NHis-R and its 5′end contains the same complementary fragment as the 3′end of the mIFc2 DNA fragment. To create the mIFc2 and LfcinB co-expression system, the mIFc2 and LfcinB gene fragments were used as templates for overlapping PCR using primer pair pET21SD1-F and LfcinB-NHis-R. A 358 bp of mIFc2-LfcinB DNA fragment, in which the 5′end contains an SpeI restriction site and independent ribosome binding site 1 (SD1), and mIFc2 and the 3′end contains the ribosome binding site 2 (SD2), LfcinB gene, and restriction sites, was amplified (Figure 3A). The stop codon of mIFc2 overlaps with the start codon of the LfcinB gene. The mIFc2-LfcinB DNA fragment was subcloned into the yT&A vector and the resultant plasmid was named pT-mIFc2-LfcinB. Using pT-mIFc2-LfcinB as a template, a 167 bp DNA fragment of fLfcinB was amplified using primers SpeI-His-F and FLfcinB-R and its 3′end contains the same complementary fragmentary as the the 5′end of the bmIFc2 (Figure 3B, lower panel). Using pT-mIFc2-LfcinB as a template, the DNA fragment (216 bp) of bmIFc2 was amplified using primers bmIFc2 and NheSmIFc2-R and its 5′end contains the same complementary fragment as the 3′end of fLfcinB (Figure 3B, lower panel). Using fLfcinB and bmIFc2 DNA fragment as template, a 370 bp of fLfcinb-bmIFc2 fusion gene fragment was amplified by overlapping PCR using SpeI-fLfcinB and NheSmIFc2-R primers (Figure 3B, lower panel). In the fLfcinB-bmIFc2 DNA fragment (Figure 3B, upper panel), the LfcinB termination codon is overlapped with the start codon from the bmIFc2. All constructs were sequenced to confirm their authenticity. Both pET21b-mIFc2-LfcinB and pET21b-fLfcinB-bmIFc2 constructs were digested with restriction enzymes BglII and SalI (New England Biolabs), respectively. The expected fragments are obtained (5432/424 bp and 5432/436 bp) (Figure 3C). The pET21b-(fLfcinB-bmIFc2)₂ plasmid DNA was digested with restriction enzymes BglII and SalI to obtain fragments of 5432 bp and 806 bp (Figure 3C). In this work, all constructs were confirmed by Sanger sequence.

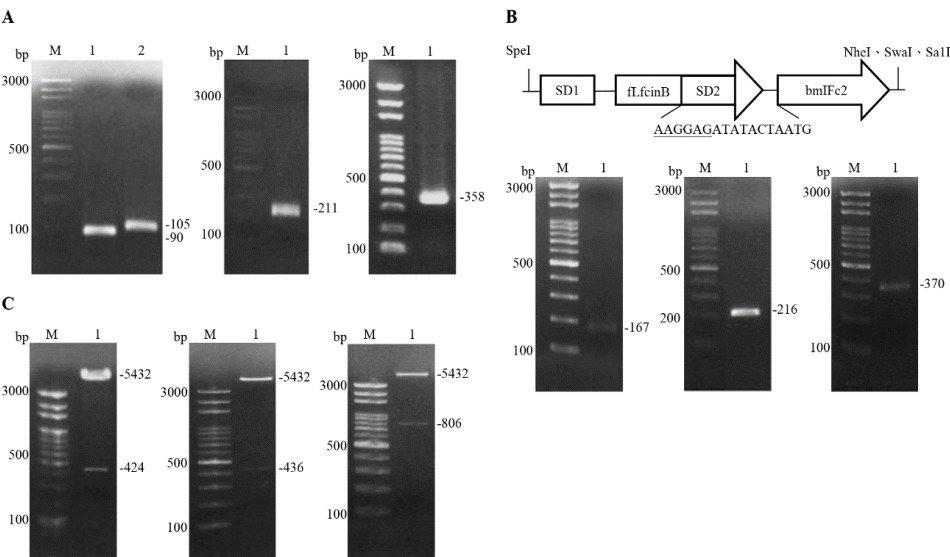

**Figure 3.** Amplification of the full-length mIFc2 DNA fragment and construction of the mIFc2 co-expression vectors. (**A**) The front mIFc2 DNA fragment (90 bp) was amplified by overlapping PCR. The rear mIFc2 DNA fragment (105 bp) was amplified by overlapping PCR using primers mIFDNA2-F and mIFDNA2-R. By using the front and rear mIFc2 DNA fragments as templates, the full-length mIFc2 DNA fragment (211 bp) was amplified by overlapping PCR. A 358 bp of mIFc2-LfcinB DNA fragment, in which the 5′end contains an SpeI restriction site and independent ribosome binding site 1 (SD1), and mIFc2 and the 3′end contains the ribosome binding site 2 (SD2), LfcinB gene, and restriction sites (NheI, SwaI, and SalI), was amplified. (**B**) The upper panel shows a gene map of the fLfcinB-bmIFc2 DNA fragment. fLfcinB termination codon is overlapped from the start codon from the bmIFc2. Using pT-mIFc2-LfcinB as a template, a 167 bp DNA fragment of fLfcinB was amplified and its 3′end contains the same complementary fragment as the 5′end of bmIFc2 (lower panel). Using pT-mIFc2-LfcinB as a template, the DNA fragment (216 bp) of bmIFc2 was amplified and its 5′end contains the same complementary fragment as the 3′ end of fLfcinB (lower panel). Using fLfcinB and bmIFc2 DNA fragment as template, a 370 bp of fLfcinb-bmIFc2 fusion gene fragment was amplified by overlapping PCR (lower panel). (**C**) Both pET21b-mIFc2-LfcinB and pET21b-fLfcinB-bmIFc2 constructs were digested with restriction enzymes BglII and SalI, respectively. The expected fragments were obtained (5432/424 bp and 5432/436 bp). The pET21b-(fLfcinB-bmIFc2)$_2$ plasmid DNA was further digested with restriction enzymes BglII and SalI to obtain fragments of 5432 bp and 806 bp.

### 3.3. Expression of the TrxA-His-SrtA-LfcinB Fusion Protein

As shown in Supplementary Figure S4, expression of LfcinB inhibited the growth of *E. coli* while co-expression of LfcinB and mIFc2 did not affect the growth of the *E. coli* host. We found that the higher expression levels of the TrxA-His-SrtA-LfcinB fusion protein and co-expression of LfcinB-His and mIFc2 were seen in *E. coli* C43(DE3). Thus, it was selected to be the host cells for later studies. To reduce LfcinB toxicity for the *E. coli* host, two different expression approaches were established to express LfcinB. In this study, the SrtA-LfcinB fusion protein was expressed in *E. coli* C43(DE3) and analyzed using a 12% SDS-PAGE and Western blot. The results show that the SrtA and TrxA-His-SrtA-LfcinB fusion protein could be expressed with the expected size of 36 kDa and 38 kDa, respectively (Figure 4A). Furthermore, the pET21b-SrtA-LfcinB construct transformed into *E. coli* C43(DE3) was induced by IPTG (200 mM) to express the SrtA-LfcinB-His fusion protein. The results show that the SrtA-LfcinB-His fusion protein could be expressed with the expected size of 21 kDa (Figure 4B). Furthermore, the expression levels of the TrxA-His-SrtA-LfcinB fusion protein by pET32-1SrtA-LfcinB, pET32-2SrtA-LfcinB, and pET32-3SrtA-LfcinB vectors in *E. coli* C43(DE3) were further analyzed by SDS-PAGE (Figure 4C). The results indicate that the higher expression levels of the SrtA-LfcinB fusion protein were obtained using the vector

with three expression cassettes (Figure 4C). The expressed TrxA-His-SrtA-LfcinB fusion protein by pET32-3SrtA-LfcinB plasmid was further purified as shown in Figure 4D. In this work, BioRad protein assay was used to analyze the concentrations and yields of proteins. Our results revealed that 500 mL of bacteria have $38 \pm 1.1$ mg of total proteins, including $22.1 \pm 1$ mg of inclusion body, and $1.32 \pm 0.07$ mg of LfcinB could be purified, accounting for about 3.5% of the total protein and 5.9% of the total inclusion body. The antibacterial activity was detected after SrtA-mediated cleavage of LfcinB in presence of $Ca^{2+}$.

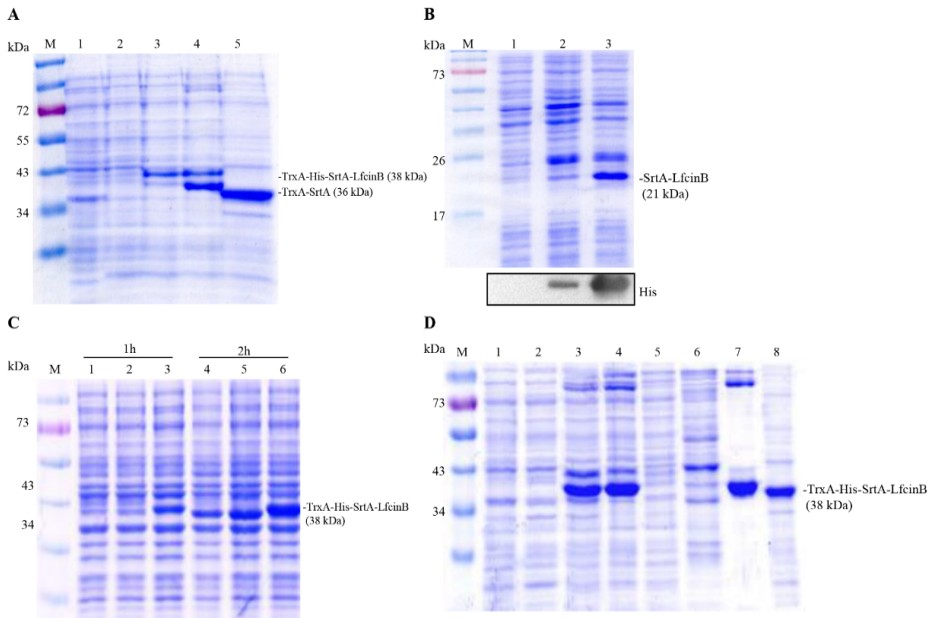

**Figure 4.** Expression and purification of the SrtA-LfcinB fusion protein. (**A**). The pET32a-SrtA-LfcinB construct was induced in *E. coli* C43(DE3) followed by induction with IPTG (200 mM) for 3 h at 30 °C. The expressed TrxA-His-SrtA and TrxA-His-SrtA-LfcinB fusion protein (38 kDa) was electrophoresed in a 12% SDS-PAGE gel. Lane M: protein marker; Lane 1: vector alone; Lane 2: without IPTG; Lane 3: IPTG induction for 1 h; Lane 4: pET32a-SrtA-LfcinB, IPTG induction for 3 h; Lane 5: pET32a-SrtA, IPTG induction for 3 h. (**B**). The pET21b-SrtA-LfcinB construct transformed into *E. coli* C43(DE3) was induced by IPTG (200 mM) for 3 h at 30 °C to express the SrtA-LfcinB-His fusion protein (21 kDa). Lane M: protein marker; Lane 1: pET21b alone; Lane 2: IPTG induction for 1 h; Lane 3: IPTG induction for 3 h. (**C**) The expression levels of the TrxA-His-SrtA-LfcinB fusion protein by pET32a-1SrtA-LfcinB, pET32a-2SrtA-LfcinB, and pET32a-3SrtA-LfcinB vectors in E. coli C43(DE3) were analyzed in a 12% SDS-PAGE gel. Proteins from the gel were transferred to a PVDF membrane. The expressed TrxA-His-SrtA-LfcinB fusion protein was detected using the His primary antibody and HRP-conjugated secondary antibody, respectively. Lane M: protein marker; Lanes 1-3: pET32 with one to three expression cassettes, induction for 1 h and 2 h (lanes 4–6). (**D**). The expressed TrxA-His-SrtA-LfcinB fusion protein using the pET32a-3SrtA-LfcinB vector was further purified by a Nickle affinity column. Lane M: protein marker; Lane 1: vector alone; Lane 2: without IPTG induction; Lanes 3-4: IPTG induction for 3 h; Lane 5: flow through; Lane 6: wash fraction; Lane 7: elution fraction; Lane 8: pET32a-SrtA, IPTG induction for 3 h.

### 3.4. Co-Expression of mIFc2 and His-LfcinB

The expression vector pET21b-mIFc2-LfcinB was transformed into E. coli C43(DE3). After induction by 200 nM IPTG, the negatively charged mIFc2 (7 kDa) and His-LfcinB (3.9 kDa) were co-expressed in insoluble form and analyzed by Western blot (Figure 5A, lanes 3, 4). BioRad protein assay was used to analyze the concentrations and yields of proteins. Our results show that 500 mL of bacteria have $41.3 \pm 1.2$ mg of total proteins, including $28.4 \pm 1$ mg of inclusion body, and $1.61 \pm 0.09$ mg of His-LfcinB could be purified, accounting for 3.9% of the total protein and 5.6% of the total inclusion body. Interestingly,

after changing the order of mIFc2 and His-LfcinB, the expression levels of His-LfcB could be further increased using the pET21b-LfcinB-mIFc2 plasmid in E. coli C43(DE3). Our results reveal that the expression levels of LfcinB using the pET21b-LfcinB-mIFc2 vector were higher than that from the pET21b-mIFc2-LfcinB vector (Figure 5A, upper and lower panels). Our data show that the total protein of 500 mL of bacteria was $43.8 \pm 1.25$ mg, including $24.2 \pm 1$ mg of inclusion bodies. In this work, $4.1 \pm 01$ mg of His-LfcinB could be purified, accounting for 9.8% of the total protein and 16.9% of the total inclusion body. Our results revealed that changing the gene order of the mIFc2 and His-LfcinB affects His-LfcinB expression levels.

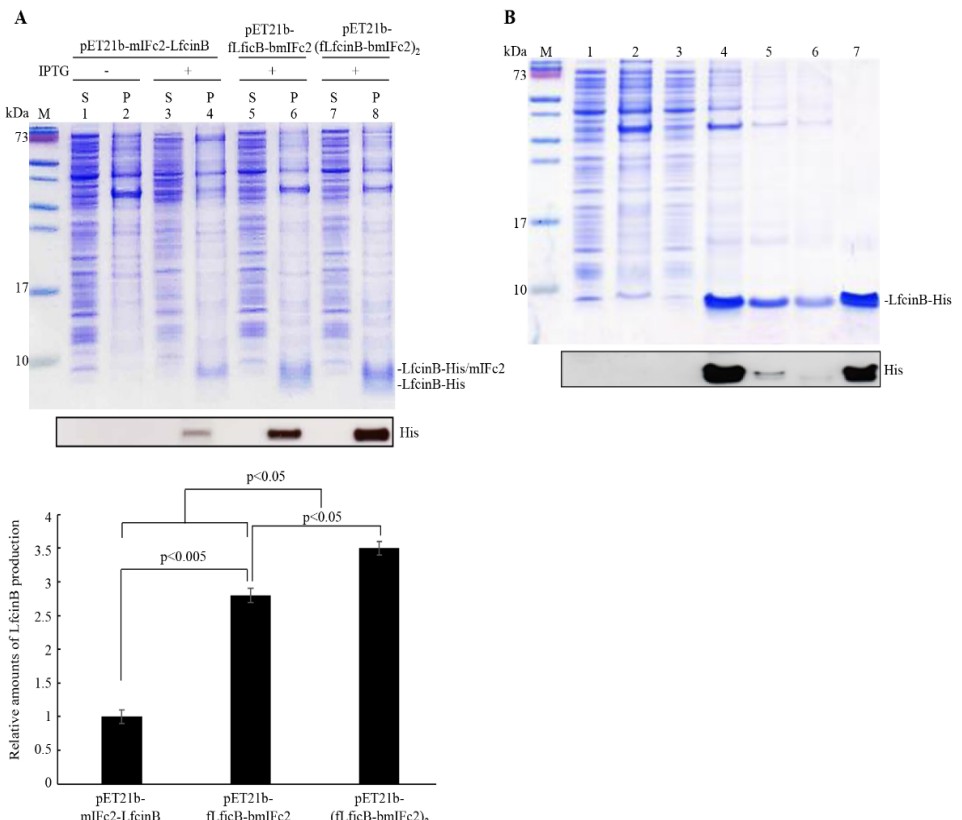

**Figure 5.** Co-expression of mIFc2 and LfcinB-His. (**A**). The expression vector pET21b-mIFc2-LfcinB was transformed into *E. coli* C43(DE3). The co-expressed proteins of mIFc2 and His-LfcinB were analyzed by Western blot with the anti-His antibody and HRP-conjugated secondary antibody, respectively. After induction by 200 nM IPTG for 3 h at 30 °C, the negatively charged mIFc2 (7 kDa) and LfcinB (3.9 kDa) were analyzed by a 16.5% tricine SDS-PAGE (upper panel). After changing the order of the mIFc2 and LfcinB, the expression levels of LfcinB-His could be further increased using the pET21b-fLfcinB-bmIFc2 plasmid. The expression levels of LfcinB by the pET21b-(fLfcinB-bmIFc2)$_2$ vector was higher than that of the pET21b-mIFc2-LfcinB and pET21b-fLfcinB-bmIFc2 vectors. The expression levels of LfcinB-His by pET21b-mIFc2-LfcinB were considered 1-fold. Signals for all blots in upper panel (lanes 4, 6, 8) were quantified using ImageJ software. All experiments were conducted in three independent experiments. (**B**) The co-expressed LfcinB-His and mIFc2 protein by the pET21b-(fLfcinB-bmIFc2)$_2$ vector was induced in E. coli C43(DE3) followed by induction with IPTG (200 mM) for 3 h at 30 °C. LfcinB-His was purified by a Nickle affinity column. Lane M: protein marker; Lane 1: without IPTG induction, supernatant; Lane 2: without IPTG induction, inclusion bodies; Lane 3: IPTG induction, supernatant; Lane 4: IPTG induction, inclusion bodies; Lane 5: flow through; Lane 6: wash fraction; Lane 7: elution fraction.

Furthermore, our results demonstrate that the production of LfcinB was significantly increased using the pET21b-(fLfcinB-bmIFc2)2 construct, which contains two copies of the

genes (Figure 5A, lower panel). The co-expressed His-LfcinB and mIFc2 proteins using the pET21b-(fLfcinB-bmIFc2)2 vector were purified and analyzed by a 16.5% tricine SDS-PAGE (Figure 5B). The expressed proteins were further analyzed by Western blot assay using the His primary antibody and HRP-conjugated anti-mouse IgG (H + L) secondary antibody (Figure 5B).

### 3.5. Enhancement of the Dissolution Rate of Inclusion Bodies by Treatments with Different Temperatures, pHs, and Resuspended Volumes

After gene replacement, higher levels of LfcinB and mIFc2 were co-expressed in the form of inclusion bodies. To increase the dissolution rate of inclusion bodies, they were treated with different temperatures and pHs and resuspended in different volumes of 50 mM Tris-HCl. First, inclusion bodies were collected and resuspended in 50 mM Tris-HCl (pH 8) followed by heat treatment in a water bath for 15 min at different temperatures (60 °C, 70 °C, 80 °C, and 90 °C). Heat-treated LfcinB/mIFc2 inclusion bodies were analyzed by a 16.5% tricine SDS-PAGE. The results show that the heat treatment at 60 °C, 70 °C, 80 °C, and 90 °C partially resolubilize the LfcinB, and heat-treated LfcinB/mIFc2 inclusion bodies at 90 °C has the best resolubilization rate (Figure 6A,B). The supernatant LfcinB obtained after 90 °C heat treatment was used to test its antimicrobial activities for Salmonella choleraesuis and Salmonella typhimurium by the agar diffusion method (Figure 6C). The result indicates that the LfcinB supernatant has antimicrobial activities.

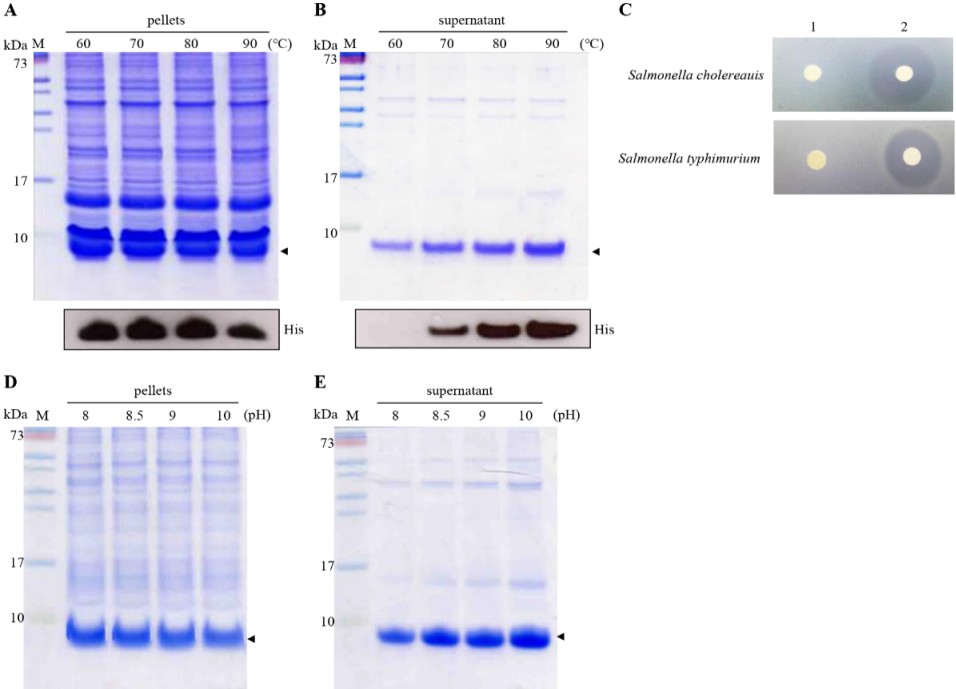

**Figure 6.** Treatments of inclusion bodies with different temperatures and pHs. (**A,B**) Inclusion bodies resuspended in 50 mM Tris-HCl (pH 8) were treated for 15 min at temperatures from 60–90 °C. Heat-treated LfcinB/mIFc2 inclusion bodies were electrophoresed in a 12% SDS-PAGE gel. Proteins from the gel were transferred to a PDVF membrane. The co-expressed LfcinB and mIFc2 proteins were detected using the His primary antibody and HRP-conjugated secondary antibody, respectively. (**C**) The supernatant LfcinB obtained after at 90 °C heat treatment was used to test its antimicrobial activities for *Salmonella choleraesuis* and *Salmonella typhimurium* by the agar diffusion method. (**D,E**) The inclusion bodies, heated for 15 min at 90 °C, were treated at pH 8 to 10. The treated LfcinB/mIFc2 inclusion bodies were electrophoresed in a 16.5% tricine SDS-PAGE. All experiments were conducted in three independent experiments.

The inclusion bodies were resuspended in 50 mM Tris-HCl at different pH (8, 8.5, 9, 9.5, and 10) heated at 90 °C for 15 min in a water bath, and the resolubilization rate of inclusion bodies was quantified by the BioRad protein assay. The pH values were 8, 8.5, 9, 9.5, and 10 and the resolubilization rates were 25.3, 35.3, 38, 44.8, and 48.1%, respectively (Figure 6D,E). The inclusion bodies were also resuspended in different volumes (1X, 2X, 4X, 6X, 8X, 10X, and 16X) of 50 mM Tris-HCl (pH 8). After heating in a water bath, quantification of inclusion bodies was performed with the BioRad protein assay. The inclusion bodies were dissolved in 1–16 times the volume of buffer, and the resolubilization rates were 25.3, 26, 30, 38.3, 47.3, 49.6, and 56.6%, respectively. Collectively, the results reveal heat-treated LfcinB and mIFc2 inclusion bodies at 90 °C, pH 10, and 16X resuspended volumes have the best resolubilization rate.

*3.6. Antibacterial Activity Tests*

In this study, an agar diffusion method was used to carry out antibacterial activity analysis. The concentration (84 ug/mL) of purified LfcinB was dropped on the indicator bacterial plate. The experimental group has an obviously bacteriostatic circle compared to the buffer alone (Figure 7). The representative results showing LfcinB with bacteriostatic ability for *Salmonella choleraesuis* (SC) and *Salmonella typhimurium* (ST) are shown in Figure 7. In addition, two-fold dilutions ranging between 18–300 mg/mL of LfcinB were prepared and their antimicrobial activities were assessed. Using the agar diffusion method and MIC test, the MIC of LfcinB on ten Gram-positive bacteria was assayed and found to be between 37 and 150 ug/mL (Table 1).

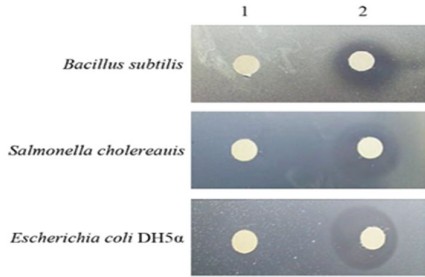

**Figure 7.** Antibacterial test using the agar diffusion method. An agar diffusion method was used to carry out antibacterial activity analysis. The concentration of 84 ug/mL purified LfcinB was dropped on the indicator bacterial plate. The experimental group has an obviously bacteriostatic circle compared to the buffer alone. The representative results showing LfcinB with bacteriostatic ability for *Salmonella choleraesuis* (SC) and *Salmonella typhimurium* (ST) are shown.

## 4. Discussion

Adding a variety of antibiotics in animal feed is a common practice in animal husbandry to prevent infectious diseases [1–5]. It is a major contributor to the increase in drug-resistant bacteria [6–9]. To overcome this circumstance, cationic AMPs show the potential to become novel antibiotics with new modes of action. LfcinB has a wide range of bactericidal activity which can inhibit Gram-positive bacteria, Gram-negative bacteria, fungi, viruses, and even parasites [34–36]. In addition to iron-binding ability, inhibition of tumor cells, and other biological functions, LfcinB has all the biological activities of bLF and has almost no toxicity to eukaryotic cells. Based on this selective effect and non-antigenic characteristics, LfcinB is expected to become a new generation of antibacterial, antiviral, and antitumor drugs. The antibacterial mechanisms of antibiotics include inhibition of bacterial cell wall synthesis, inhibition of protein synthesis, inhibition of nucleic acid synthesis, and inhibition of the synthesis of important metabolites [37,38]. It is mainly to reduce the permeability of the cell membrane to antibiotics. The main antibacterial mechanism is to collapse the cell membrane to achieve the antibacterial effect. This mechanism is different from the mechanism of antibiotics that inhibit the biological activity of bacteria, such as inhibition of cell wall production or inhibition of DNA and RNA synthesis. It is not easy to

develop drug resistance due to mutations, and it is possible to develop drug resistance unless the cell membrane composition is changed. In addition, using the structural properties of antibacterial peptides can help in the design of new antibacterial peptides [39–41], and even to change the amino acid composition of antibacterial peptides. It can also increase the antibacterial ability or reduce hemolysis. Therefore, it may be used as a new generation of antibiotics and has the potential to replace the existing antibiotic system.

Interest in the production of AMPs for pathogens is growing. The most commonly used method in previous studies has been fusion of an antibacterial peptide with a protein that is not toxic to *E. coli* to reduce its toxicity to bacteria. Several reports have shown that enzymes (such as furin) and chemicals (such as CNBr) are used for protein cleavage to release the active antimicrobial peptides [28,29]. However, this production method is limited to research purposes, because the fusion proteins need to undergo a purification step and a cleavage step to obtain the target antibacterial peptide, and the yield of the antibacterial peptide released by cleavage is low. In this work we have developed and compared two expression approaches to address this issue. Among them, the SrtA fusion expression system is specifically improved for fusion protein cleavage. The fusion protein is designed as a protein with protease activity, and the step of adding enzymes or chemicals can be omitted through auto-cleavage. After expression of LfcinB, the enzyme activity of SrtA can be used to induce self-cleavage and release of antibacterial peptides [30]. The advantage of this expression system is that SrtA is a soluble protein, and the TrxA-His-SrtA-LfcinB fusion protein is soluble and neutralizes the toxicity of the antimicrobial peptide, making it suitable for expression in *E. coli*. However, this expression system still has some problems to be overcome. For example, after the expression of the fusion protein, some autologous cleavage occurs inside the host, causing a small amount of LfcinB to be released to inhibit the growth of the *E. coli* host, which affects the expression of LfcinB. Thus, the LfcinB and mIFc2 co-expression system was aimed at improving the shortcomings of the SrtA fusion protein expression system. The principle of the mIFc2 antibacterial peptide co-expression system is to mutate the hIFN-γ sequences into a negatively charged mIFc2 protein, which has hydrophobic characteristics and is compatible with antibacterial peptides [32]. After co-expression, the negatively charged mIFc2 are polymerized with LfcinB into insoluble inclusion bodies. The negative charge of mIFc2 neutralizes the positive charge of LfcinB to reduce its toxicity to *E. coli*. Its advantages are that it can directly express LfcinB without being a fusion protein, it does not require protein cleavage steps, and it has a higher protein yield. Our results reveal that its yield can reach 9–10% of the total protein. A previous report suggested that inclusion bodies could be denatured by the use of urea and re-dissolved in a buffer solution. It can be further purified through a cation exchange resin based on the positive charge of the antibacterial peptide.

A previous study pointed out that soluble and active antimicrobial peptides can be directly recovered by a heat treatment temperature of 100 °C, indicating that they have thermal stability [42]. Therefore, we used heat treatment to destroy the interaction between LfcinB and mIFc2 to obtain the solubilized monomeric LfcinB. Our data revealed that heat-treated LfcinB/mIFc2 inclusion bodies at 90 °C have the best resolubilization rate. Our results suggest that heat-treated LfcinB and mIFc2 inclusion bodies at 90 °C, pH 10, and 16X resuspended volumes have the best resolubilization rate. It is possible that some negatively charged mIFc2 are included; however, it can also be purified by cation exchange resin to obtain pure monomeric LfcinB. In the future, the conditions of heat treatment and purification process can be adjusted to improve the release rate of Lfcin B from the inclusion body polymer, so as to increase the yield and reduce the cost. In this work, we show that LfcinB co-expressed with mIFc2 exhibits excellent broad-spectrum antibacterial activities against Gram-negative and Gram-positive bacteria species with a range of MIC between 37–150 ug/mL. Comparing these two expression systems, the mIFc2 co-expression system shows higher efficiency for LfcinB production than the SrtA fusion system, suggesting that this co-expression system has better practical potential.

Compared to previous studies [43], the MIC of LfcinB for *Salmonella choleraesuis*, *Escherichia coli*, and *Bacillus subtilis* was 37 ug/mL, 100 ug/mL, and 75 ug/mL, respectively. The lowest inhibitory concentration was with *Salmonella choleraesuis*, and the results were in line with expectations. The experimental concentration data for *Escherichia coli* and *Bacillus subtilis* were high. This may be due to the insufficient purity of the LfcinB. Compared with ferritin, the concentration is overestimated; another reason is that the different strains used in the experiment may also cause the MIC to be different. Previous studies pointed out that there will be a synergistic effect between antibacterial peptides, and different antibacterial peptides working together will increase the antibacterial activity [43]. Synergy also occurs between antibacterial peptides and antibiotics [44]; when LfcinB acts simultaneously with some antibiotics, it can increase the effectiveness of other antibacterial substances and increase its antibacterial effect on resistant bacteria. When it has a synergistic effect with erythromycin, minocycline, and monoacylglycerol, it can enhance the toxic ability on *E. coli* and drug-resistant *Staphylococcus aureus* [45], so it can be used in animal husbandry with other antibiotics or antibacterial peptides. In addition, the advantage of bovine lactoferrin is its safety as a feed or food additive. When the calf has not developed an acquired immune system, it is in the milk released by pepsin cutting ferritin and is an important substance for innate immunity [46]. Thus, it could be used as a food additive.

**Supplementary Materials:** The following supporting information can be downloaded at: https://www.mdpi.com/article/10.3390/pr10122470/s1, Supplementary Table S1 and Supplementary Figures S1–S4.

**Author Contributions:** All authors made substantive intellectual contributions to the present study and approved the final manuscript. H.-J.L. conceived the study and generated the original hypothesis, wrote the paper, and supervised the project; C.-Y.H. (Chao-Yu Hsu) and C.-Y.H. (Chung-Yiu Hsieh). performed most of the experiments. C.-Y.H., C.-Y.H., C.-Y.Y., Y.-K.C., W.-L.S., C.-M.Y., N.-J.H., M.-S.C., B.L.N. and H.-J.L. analyzed data; H.-J.L. and B.L.N. All authors have read and agreed to the published version of the manuscript.

**Funding:** This work was supported by grants from Tungs' Taichung MetroHarbor Hospital (TTMHH-R1100025), Taichung Veterans General Hospital and National Chung Hsing University (TCVGH-NCHU1117608), the iEGG and Animal Biotechnology Center from the Feature Areas Research Center Program within the framework of the Higher Education Sprout Project of the Ministry of Education (MOE) in Taiwan (111S0023A), and the Ministry of Science and Technology in Taiwan (109-2313-B-005-006-MY3& 111-2622-B-005-001).

**Institutional Review Board Statement:** Not applicable.

**Informed Consent Statement:** Not applicable.

**Data Availability Statement:** Not applicable.

**Conflicts of Interest:** We declare that we have no competing interests.

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
