# Peer review of "Sortase A Fusion Expression and mIFc2 Co-Expression of Bovine Lactoferricin and Analysis of Its Antibacterial Activity"

_processes, doi:10.3390/pr10122470_

Round 1
Reviewer 1 Report
In this manuscript, the authors describe a recombinant method for the production of lactoferrins in E. coli heterologous expression system. This is an interesting and challenging topic as often the expression of AMPs in bacteria results in poor yields and induce toxicities. However, there are several areas in this manuscript that needs significant improvements. Therefore, I am of the opinion that the manuscript can only be considered after some major revisions. Major Comments:
1) It is not clear whether the fusion construct or the co-expression with an anionic peptide actually improves the peptide expression. The authors need to compare the expression levels with a non-fusion lactoferrin B expression cassette . I suggest the authors to use the 6XHis Lfcin B expression cassette as control to compare the expression levels and toxicities with the fusion construct and co-expression constructs.
2) Since it is method/construct development, the authors need to provide the purified protein obtained for each construct and expression conditions in a quantitative manner. While SDS-gel images are provide a qualitative information about the expression levels, which is not sufficient. Also, the authors need to provide the amount of purified protein obtained from at least three independent expression experiments and values should be provided as mean and Stdev.
3) The authors have used heat treatment method to purify peptides from inclusion bodies. Lactoferrin B has two Cys ( Page 2 of the manuscript). Therefore, there is a potential possibility that in the inclusion body, peptides form a higher order oligomers linked through disulfide bonds. When dissolving the inclusion bodies, the authors are supposed to add a reducing agent and do subsequent refolding of the peptide.
4) The authors need to include a synthetic lactoferrin B peptide as standard to compare the antimicrobial activity under their testing conditions for the comparison and show unequivocally that peptide has similar antibacterial profile as synthetic peptide.
5) The authors have used SDS-PAGE to quantify the amount of soluble lactoferrin B from inclusion bodies after heat-treated dissolution method. The authors should use Native-PAGE for the same. Heating the sample in the presence of SDS will dissolve the oligomers ( formed through non-covalent interactions) present in the soluble fractions to monomers, can overestimate the amount of monomers. Also, I suggest the authors to explicitly state whether the electrophoresis where done under reducing or non-reducing conditions.
Minor Comments:
1) The authors are encouraged to reduce the length of the paper, some of the cloning strategies can be moved from the main text and provided as Supplementary data.
2) The authors need to consider their statement "In this work, we found that E. coli C43(DE3) has better anti-toxin ability, it was selected as host cells for latter studies" given on page 13, as there is no experimental evidence to prove the same.
3) The authors are encouraged to provide a higher quality images for the disc-diffusion assays.
4) On Fig 10, the clear zone observed for the peptide treated discs appear to asymmetric, it would be interesting to know why such asymmetry is observed.
Author Response
Please the attached file

Reviewer 2 Report
The manuscript “The Sortase A Fusion Expression and mIF c2 Co-Expression of Bovine Lactoferricin and analysis of its Antibacterial Activity” presented new method by fusing SrtA to lactoferricin using standard molecular cloning approaches to create two different constructs to express LfcinB. That said, the reviewer find that the manuscript requires major edits in English as it is written. The description of methods was lengthy and repetitive. The introduction and discussion lacks references. Results presentation require rearrangement of figures. Some figures can be in supplemental and does not need to be in the actual report.
Consider the following comments below:
Introduction:
Page2: Reword/rework the first 2 paragraph. You mention of animal feed additives and antibiotics, please give examples and references. You mentioned antibiotic resistance being developed as human, please list some references such as those from CDC (USA), WHO, CDC (EU), and CDC (TW) if any. You mentioned innate immune system, what is it that you speak of? Human? Animal? It is not clear and please list references. You speak of AMP does not develop resistance easily? Please list examples and references.
It would be helpful if you could have a figure that shows the amino acid alignments of Both LfcinH and LfcinB to show the similarities and differences and list references. The introduction would greatly benefit from briefly talk about why you choose the hybrid/fusion methods and what advantages you would gain, and what other people have done in the past. Have people tried such method to express other non-toxic/Amp peptides? Also briefly mention SrtA (even you did at the discussion). Also the type of bacteria that you choose to test, why 13 G- and 10 G+? why these bacteria?
Materials and Methods:
Page 3: Why you choose to test these E. coli expression strains? What about Acella? KRX or Rosetta?
Page 5 – 6 figures: The construction of your vector description can be more concise. Why not just synthesize your genes? Rather than cloning piece by piece?
Page 7: Repetitive in your description of IPTG. Please also list all your chemicals vendors including your tricine gels. Your construct map did not show his-tag in Figure 2 and 3. List your Nickle resin (Ni-NTA or?) also company you bought from. List your antibody used here (concentration and type of antibody. Anti-His-tag?) and company you bought from.
Page 7 – 8: Agar diffusion and MIC was too wordy and not enough detail provided. No do to say it is “easy to perform, simple method…” You also need to include reference of your method that you are using. 96-well plate is it flat or round bottom? Statistical analysis was done with what software package? MS Excel or Sigma Plot or …
Page 9 – 12: Figure 4, 5, 6, 7. I think some of these figures can be put into supplemental materials and does not need to be included in the actual manuscript. Similarly, figure 8 and 9 in page 14 and 15. Figure 10 in page 16 needed to be resized properly.
Page 17 – 18:
Discussion: Please see some of my comments on introduction. You need more detail and example on the antibiotic usage in farm animal feed. Also, unsure what you mean by “…cell membrane composition is changed”. Please be more specific and provide references.
Have you thought about using other fusion protein? Such as enterokinase?
Page 18: …”no doubt about its safety as food additive.” Unless you test it, it is not advised to say “no doubt”.
Reviewer 3 Report
Major comments
The article describes two strategies to improve the expression of the AMP LfcinB in E. coli. One in fusion with SrtA, to obtain a soluble AMP, and the other coexpression with mIFc2 to obtain inclusion bodies. The recombinant LfcinB was tested against many bacteria.
The article is on the scope to publish in this journal because it compared two different approaches to express an AMP in bacteria and showed that the order of sequences mIFc2 and LfcinB in the vector is important. However, the work must be improved.
The objectives of the work were not fully met to justify the publication of the work in that way. Both strategies could be better addressed; only the amount of purified peptide does not justify the publication.
The LfcinB obtained after the cleavage of the fusion peptide could be evaluated since amino acids added due to the cloning strategy may interfere with the antibacterial activity. It is interesting to test both recombinants LfcinB, the soluble and the insoluble, to infer the antibacterial activity.
Did the peptides used in the antibacterial tests present S-S bridges? The structure of both recombinants is the same? Include the purification of LfcinB after proteolysis with SrtA. The efficiency yield of both processes was not shown, ug of peptide / total protein extract after lysis.
An accurate spell check and revision of the manuscript are required.
Homogenize the name of the plasmids resulting from the pET32SrtA-LfcinB cassettes in the text.
Minor comments
Introduction
It is necessary to improve the Introduction; much research on LfcinB illustrates the interest in this AMP.
It could give more information about SrtA and mIFc2 to show their sizes in the construction of expression vectors; alternatively, this information could be provided in Methods.
Material and methods
What templates are used for amplifying SrtA, LfcinB, and mIFc2 sequences? It was not evident in the text.
Correct the references in item 2.3: LfcinB is [11], is replaced with SrtA, and the article [15] is not about SrtA protease.
Results
As a result of the plasmid constructions, present only the sequences of the expression cassettes of pET32-(SrtA-LfcinB)1, pET32-(SrtA-LfcinB)2, and pET32-(SrtA-LfcinB)3 as also of pET21b-(fLfciBbmIFc2)2, indicating the peptides that will be expressed, the fusion and respective cleavage site and other pertinent information. It is unnecessary to present the gels of PCR and digestion with restriction enzymes of vectors.
Fig. 2, I did not understand why the cassette of pET32-SrtA-LfcinB has 1100 bp if the fusion gene has 546 bp.
Fig. 4, Place the B at the correct place, the gel with the 92 bp amplification.
Place the C at the correct place, the gel, with the 546 bp amplification.
Include item D in the legend and add that to the digestion gel.
Fig. 5, indicates the restriction enzymes used below the lane.
Fig. 6, has a legend that is too hard to read. Try to facilitate the visualization of data.
Fig. 7, I did not understand why the fusion of pET32a-SrtA-LfcinB has 38 kDa and the fusion protein expressed by pET21 has 21 kDa. Correct the size of SrtA-LfcinB of pET32a-(SrtA-LfcinB)3 in lane 8, it hasn’t 32 kDa.
It is unnecessary to present the gels A and B, but it is interesting to show the western blot of the fusion protein of pET32a-(SrtA-LfcinB)3.
Fig. 8, Include chart information in the figure legend.
A suggestion to the legend: A.B. “The inclusion bodies, resuspended in 50 mM Tris.HCl pH 8, were treated for 15 min at temperatures from 60-90 °C. D.E. The inclusion bodies, heated for 15 min at 90 °C, were treated at pH 8 to 10.”
Fig. 9, A suggestion of title "Treatments of inclusion bodies with different temperatures and pHs.
Change item C to Fig. 10.
Indicate pellets and supernatant in gels A and B, as done to gels D and E.
3.7. A suggestion of title "Antibacterial activity tests", MIC also is an antibacterial test.
Fig. 10, A suggestion of title “Antibacterial test using the agar diffusion method”.
What is the volume dropped on the plate?
Table 2. A suggestion of title "Minimum Inhibitory Concentrations of LfcinB determined using microbroth dilution method".
Discussion
Improve the discussion.
The first paragraph could be used in the Introduction because it is not a discussion but a justification for using LfcinB.
There are no results to affirm this: "Second, although the antibacterial peptides released by this expression system are active, they are of little practical value due to low yield". The authors did not present the antibacterial activity of LfcinB derived from fusion peptide nor the yield of this strategy.
Why insufficient purity decreases the activity against Gram-positive bacteria and not against Gram-negative bacteria?
Are there other strategies to improve the recombinant Lfcin than the synergism?
Round 2
Reviewer 1 Report
The authors have tried to address some of my concerns. However, the authors still need to make sure that the peptides isolated from the inclusion bodies are monomeric in nature. As I have mentioned previously, the authors need to run a Native-PAGE under non-reducing conditions to confirm the solubilized peptide as monomeric lactoferrinB. Alternatively, they can also run LC-MS analysis to confirm the peptide as lactoferrin B.
Author Response
NA

Reviewer 2 Report
Authors had made substantial improvement in addressing reviewer’s previous comments. However, there are still areas that need to be addressed. Please consider the following:
Abstract: Consider rephrasing the first paragraph as it is disjointed.
Introduction:
Edit “-Im-proper” use of antibiotics-and …
Provide policy/reference to “some countries banning the use of antibiotics”
Materials and methods:
2.1: What company/source did you obtain your E. coli ?
Table 1: Consider adding a reference/source column to your bacteria strains and add your E. coli strains into Table 1 as well.
2.4: Similarly, list a table of all the plasmids that you’ve used or created with reference/source.
What about the source/company that you purchased your restriction enzymes from? This was not addressed from previous review.
Edit: Last sentence. “All developed vectors were sequenced to confirm their authenticity” consider “All plasmids were confirmed by Sanger sequence. “
2.5: Remove this entire section as it is redundant and there is no need to describe how to conduct PCR as it is considered as a basic molecular cloning technique. Alternatively, you can consolidate this section into 2.3.
2.6: Again, list your chemicals source/company that you purchased from. If they are all from the same company (i.e., Sigma) just state it.
2.7: Edit: “100 uL” to “100 µL” List your sonication parameter or settings. Also list your sonicator brand/model.
Dilution factor usually is worded as (1:1,000 ratio) consider revise.
“3000” to “3,000” and “5000” to “5,000”
2.9: “pH (8, 8.5, 9, 9.5, and10)” add space “and 10”
2.10: “The agar dilution” to “diffusion”
How do you determine growth of bacteria as stated “…the growth of bacteria is determined.” By OD? or McFarland?
Edit “50 mM Tris-HCl” to Fifty millimolar Tris-HCl …
3. Results:
Figure 2. Edit: “The expected size of the pCR” to “PCR”
3.1 – 3.3 are considered method and should be moved to section 2 materials and methods. Results starts with 3.4.
4 Discussion:
Based on your introduction, the goal is to reduce toxicity to E. coli when expressing LfcinB, you have not really talked about your measurement of E. coli toxicity. Consider expanding a few sentences to make this point clear beside increasing protein expression level. To what level is it less toxic compared to other systems? 10 – 15%?
Author Response
NA
Reviewer 3 Report
The manuscript remains difficult to read, despite the change of the vector construction figures to the Supplementary Material. I suggest that the entire description be done together with the results in the supplementary material and that the sequencing of both the fusion and co-expression cassettes be presented as a result in the main text, as already suggested in the first review. Due to the fusion purification efficiency presented by the authors (about 3X lower than the co-expression), it is possible to test the antimicrobial activity of the LcfinB recovered in the fusion strategy. This would greatly improve the article, as already suggested in the first review. The discussion should be improved concerning the work results, including the process of purification.
Author Response
NA
